# White Matter Correlates of Early-Onset Bipolar Illness and Predictors of One-Year Recurrence of Depression in Adults with Bipolar Disorder

**DOI:** 10.3390/jcm11123432

**Published:** 2022-06-15

**Authors:** João Paulo Lima Santos, Michele Bertocci, Genna Bebko, Tina Goldstein, Tae Kim, Satish Iyengar, Lisa Bonar, MaryKay Gill, John Merranko, Anastasia Yendiki, Boris Birmaher, Mary L. Phillips, Amelia Versace

**Affiliations:** 1Department of Psychiatry, University of Pittsburgh, Pittsburgh, PA 15260, USA; bertoccima@upmc.edu (M.B.); bebkog@upmc.edu (G.B.); goldtr@upmc.edu (T.G.); ssi@pitt.edu (S.I.); bonarlk@upmc.edu (L.B.); gillmk@upmc.edu (M.G.); merrankoja@upmc.edu (J.M.); birmaherb@upmc.edu (B.B.); phillipsml@upmc.edu (M.L.P.); versacea@upmc.edu (A.V.); 2Magnetic Resonance Research Center, Department of Radiology, University of Pittsburgh, Pittsburgh, PA 15260, USA; tak19@pitt.edu; 3Athinoula A. Martinos Center for Biomedical Imaging, Department of Radiology, Massachusetts General Hospital, Harvard Medical School, Boston, MA 02129, USA; ayendiki@mgh.harvard.edu

**Keywords:** bipolar disorder, early-onset, recurrence, depressive episodes, neural predictors, dMRI

## Abstract

Diffusion Magnetic Resonance Imaging (dMRI) studies have reported abnormalities in emotion regulation circuits in BD; however, no study has examined the contribution of previous illness on these mechanisms. Using global probabilistic tractography, we aimed to identify neural correlates of previous BD illness and the extent to which these can help predict one-year recurrence of depressive episodes. dMRI data were collected in 70 adults with early-onset BD who were clinically followed for up to 18 years and 39 healthy controls. Higher number of depressive episodes during childhood/adolescence and higher percentage of time with syndromic depression during longitudinal follow-up was associated with lower fractional anisotropy (FA) in focal regions of the forceps minor (left, F = 4.4, *p* = 0.003; right, F = 3.1, *p* = 0.021) and anterior cingulum bundle (left, F = 4.7, *p* = 0.002; right, F = 7.0, *p* < 0.001). Lower FA in these regions was also associated with higher depressive and anxiety symptoms at scan. Remarkably, those having higher FA in the right cluster of the forceps minor (AOR = 0.43, *p* = 0.017) and in a cluster of the posterior cingulum bundle (right, AOR = 0.50, *p* = 0.032) were protected against the recurrence of depressive episodes. Previous depressive symptomatology may cause neurodegenerative effects in the forceps minor that are associated with worsening of BD symptomatology in subsequent years. Abnormalities in the posterior cingulum may also play a role.

## 1. Introduction

Bipolar Disorder (BD) is a chronic mental illness associated with high rates of psychosocial impairment, morbidity, and mortality, and represents one of the most debilitating illnesses worldwide [1]. Characterized by mood fluctuations between depression, mania, and/or hypomania, BD is a lifelong condition typically diagnosed during adolescence and young adulthood [2]. Early-onset BD has been associated with increased suicidality, functional impairment, and worse clinical trajectories [3,4]. Psychiatric comorbidity is common in BD, with high prevalence of anxiety and substance use disorders compared to the general population [4]. Impulsivity (e.g., risk-taking behaviors) has been identified as an important predictor of BD illness severity and substance use comorbidity [5]. However, little is known about the neural correlates of having experienced BD symptoms from an early age and whether they are associated with clinical worsening over the years.

Over the past decades, the neural mechanisms underlying BD psychopathology have been broadly investigated [6,7,8]. Diffusion Magnetic Resonance Imaging (dMRI) has been used to show micro- and macro-structural properties of white matter tracts connecting key functional brain regions [9]. Fractional anisotropy (FA), in particular, represents an index of water directionality in a given tissue and reflects the collinearity and/or integrity of the fibers [9]. Lower FA has been consistently reported in key white matter tracts involved in emotional processing and regulation in youth and adults with BD when compared to healthy controls [10,11,12,13,14,15,16,17,18,19,20]. Specifically, lower FA in the uncinate fasciculus (UNC) [10,11,12,13,14,15,16,17], cingulum bundle (CB) [13,17,18,19], and forceps minor (FMIN) [13,20] has been associated with the pathophysiology of BD. Yet, it remains unclear whether dMRI abnormalities in one or more of these tracts represent the effect of previous BD symptomatology or can further contribute to the prediction of recurrence of mood episodes. Elucidating the relationship between previous BD symptomatology, white matter integrity, and recurrence of mood episodes will further our understanding of the pathophysiological mechanisms of BD.

Using a combination of global probabilistic tractography, a tract-profile approach, and a feature selection model, the aims of the present study were to identify neural correlates of previous (pre-scan) BD symptomatology and further investigate the extent to which these or other dMRI measures can help predict recurrence of (post-scan) mood episodes. The use of a tract-profile approach allows us to acknowledge that the diffusion properties are not uniform along the tracts [21] and to inform specific regions of the tracts affected by early-onset BD. Leveraging the Course and Outcome of Bipolar Youth (COBY) study, a project that has been instrumental for the identification of demographic and clinical predictors (e.g., earlier age of onset, severity of illness) of different BD courses [22,23,24], dMRI data were collected in BD individuals who have been clinically followed up for up to 18 years. In addition, participants were clinically followed for one additional year after the scan to identify recurrence of mood episodes. During this time, incidence of (hypo)manic episodes was infrequent and therefore the post-scan study focused on recurrence of depressive episodes.

Based on the extant BD literature [10,11,12,13,14,15,16,17,18,19,25,26,27], we hypothesized that higher frequency of (pre-scan) mood symptoms in BD participants would be associated with lower fiber collinearity in emotional regulation tracts (FMIN, CB, and UNC; primary hypothesis). As noted above, no previous study examined the extent to which neural correlates of previous BD symptomatology can help predict recurrences in BD; however, we hypothesized that abnormalities in some of these white mater tracts would be associated with the recurrence of depressive episodes one year after scan, differentiating BD participants with post-scan depressive episodes from both those with no recurrence and healthy controls (secondary hypothesis). 

## 2. Materials and Methods

### 2.1. Participants

The study obtained approval from the Office of Research Protections at the University of Pittsburgh. Seventy-eight BD adults (mean age (SD) = 26.5 (3.9); age range = 19–33 years old; 48.7% female, 73.1% Caucasian), who were recruited through COBY when they were on average 12.1 (3.1) years old, were included in this study (R01MH059929; PIs: Birmaher, Phillips, Versace). Eight participants (10.2%) were excluded from the analysis due to incomplete neuroimaging acquisition or poor-quality scans, leaving a total of 70 BD participants (mean age (SD) = 26.3 (3.9); age range = 19–33 years old; 50.0% female, 74.3% Caucasian). Thirty-nine age, sex, and handedness group matched adults with no Axis-I diagnoses (HC) were recruited through an ongoing neuroimaging study, namely the Dimensions of Affect, Mood, and Neural Activity Associated with Distress (DIAMOND) study (R01MH100041, Phillips) and a local recruitment website (https://pittplusme.org), and served as reference for dMRI measures. Axis-I diagnoses were excluded by trained clinicians that administered the SCID-5-RV (Research Version) [28].

Exclusion criteria are detailed in the Appendix A. Demographic and clinical characteristics of the sample are in Table 1 and Appendix A. 

### 2.2. Clinical Assessments 

The Longitudinal Interval Follow-up Evaluation (LIFE) [29] and Psychiatric Status Ratings (PSRs) [29] were used to rate severity and frequency of depression, mania, and hypomania for up to 18 years before scan (range = 11.3–18.4 years, mean (SD) = 14.4 (1.6) years), and for one year after scan in BD participants (Figure 1). Follow-ups were scheduled approximately every year (average interval (SD) = 11.52 (8.3) months).

The LIFE has been validated and extensively used to ascertain longitudinal clinical data [30,31,32]. This instrument evaluates the course of symptoms by identifying change points, frequently anchored by memorable dates for the participant (e.g., holidays and beginning of school). The interviewer first reviewed with the participant how they were doing at the time of the last assessment regarding any symptoms. Then change points were assessed (e.g., “Did things get worse or better since the last time? When did things change?”) and once a change point was established, symptoms were reassessed to determine level of severity. To help the participant date these change points, the interviewer asked questions such as “Was that in November?”, “Did that happen before or after Christmas?”, etc. The interviewer then tracked the PSR for these change points forward to the current visit, probing until the best level of recovery was determined. Although these ratings were made on a week-by-week basis, participants did not have to specify how they were feeling during each week. Instead, weekly ratings were based on these change points. This process was repeated for any change points (symptoms worsening or improving). At intake, youth and parents were interviewed to evaluate the presence of current and lifetime psychiatric disorders. At each follow-up, the LIFE was administered to youth and parents (to provide collateral information). Usually, children were interviewed with their parents and adolescents and their parents were assessed separately. Any discrepancies between the informants’ responses were discussed, and a summary score based on all available information was determined [24]. 

The PSR score used to rate depression, mania, and hypomania symptoms ranged from 1 to 6 with 1–2 being no or minimal symptoms, 3–4 subsyndromal symptoms, and 5–6 full Diagnostic and Statistical Manual of Mental Disorders (DSM) IV criteria for syndromal episodes (Appendix A). 

Assessments were conducted by research staff trained to reliably administer the interviews. Psychiatrists or psychologists confirmed all diagnoses and PSRs. Up to six interviewers rated the same interview and coded the PSR for the time interval assessed during the interview. Ratings for the presence and severity of mood episodes and comorbid disorders were compared. Intraclass correlation coefficients (ICCs) were estimated for continuous variables (e.g., percent of weeks with threshold symptoms) and Kappas were estimated for nominal variables (e.g., presence vs. absence of symptoms). For this study, the overall KSADS Kappa coefficients for psychiatric disorders were ≥0.8. The ICC for the KSADS depression section was ≥0.95. The ICC for syndromal and sub-syndromal episodes ascertained through the PSR scales (using methods described elsewhere [33]) were ≥0.75; the ICC and the Kendall’s coefficients of concordance were between 0.74 and 0.79 for a major depressive episode and between 0.60 and 0.67 for mania/hypomania.

Longitudinal measures of interest were divided into (1) pre-scan and (2) post-scan.

#### 2.2.1. Pre-Scan Longitudinal Variables

Based on the weekly information about mood symptoms, seven measures were calculated representing the percentage of time experiencing (1) syndromic depression, (2) syndromic mania, (3) syndromic hypomania, (4) sub-syndromic depression, (5) sub-syndromic hypomania, (6) mixed symptoms, and (7) euthymia. Notably, mania or hypomania classification was mutually exclusive—presenting with one automatically excluded the other, and sub-syndromic symptoms of mania were classified as sub-syndromic hypomania. Mixed symptoms represented periods of time in which participants experienced syndromic or sub-syndromic symptoms for both depression and mania/hypomania. These measures were not correlated with the time between follow-ups (*p* > 0.05 for all correlations). In addition, four measures were calculated representing the percentage of follow-up time taking the following medications: antidepressants, antipsychotics, lithium, and non-lithium mood stabilizers. Age of BD onset, socioeconomic status at study entry, and number of mood episodes (depression, mania, hypomania, and mixed) before study entry (during childhood/adolescence) were also collected. 

Pre-scan longitudinal data were available for all 70 BD participants. 

#### 2.2.2. Post-Scan Longitudinal Variables

Based on weekly information about mood symptoms after scan, presence of depressive, manic, and hypomanic episodes was evaluated. Depressive episodes were defined as two or more consecutive weeks of PSR ≥ 5 for depression. Manic and hypomanic episodes were defined as one or more weeks with PSR ≥ 5 for a mania and hypomania respectively. 

Post-scan longitudinal data were available for 52 BD participants (mean age (SD) = 25.9 (3.8); age range = 19–33 years old; 51.9% female, 73.1% Caucasian). All 52 BD participants were in full or partial remission at-scan (PSRs < 5 for at least 2 months before scan). For additional information, see Appendix A. Nineteen BD participants (36.5%) showed a depressive episode during post-scan longitudinal follow-up. Only one participant showed a manic episode, and none showed hypomanic episode during post-scan follow-up. 

Based on the recurrence of depressive episodes post-scan, two BD groups were derived: (1) BD with depressive episodes (Depressed BD, N = 19), and (2) BD participants without depressive episodes (Non-depressed BD, N = 33). 

#### 2.2.3. Additional Clinical and Demographic Measures

At the time of the scan, all participants were interviewed using the Hamilton Depression Rating Scale (HDRS) [34] and the Young Mania Rating Scale (YMRS) [35]. Furthermore, all participants completed the Affective Lability Scale (ALS) [36], Barratt Impulsiveness Scale (BIS) [37], Sensation Seeking Scale (SSS) [38], State-Trait Anxiety Inventory (STAI Form Y) State and Trait [39], and 90 items-Mood and Anxiety Symptom Questionnaire (MASQ90—Anhedonic Depression, Anxious Arousal, Loss of Interest, and General Distress-Depressive, Anxious, and Mixed) [40]. 

Information regarding demographics, comorbid psychiatric disorders (anxiety, psychotic, personality, developmental, substance use, and ADHD) and pharmacological treatment (antidepressants, antipsychotics, and mood stabilizers) were also collected at scan (for additional information, see Appendix A).

### 2.3. Neuroimaging Data

The steps for acquisition and preprocessing of diffusion-weighted images are described in the Appendix A. Reconstruction of five major white matter tracts implicated in emotion regulation (FMINOR, Left/Right CB, and Left/Right UNC) was performed using TRActs Constrained by UnderLying Anatomy (TRACULA) [41] in the FreeSurfer [42] package. For each tract, an overall mean FA and nodal FA values were extracted to depict the collinearity of the fibers across (mean) and along (tract-profile) the entire tract. Tract-profile analyses allow for the characterization of white matter micro-structural properties along the tract (100 consecutive nodes) and thus for the identification of focal abnormalities in tracts of interest. Other tracts (forceps major, left/right anterior thalamic radiation, inferior longitudinal fasciculus, superior longitudinal fasciculus-temporal, superior longitudinal fasciculus-parietal, and corticospinal tract) were also reconstructed to explore the contribution of previous symptomatology on other major white matter tracts. Axial and Radial Diffusivity (AD and RD, respectively) were also extracted as they reflect the displacement of water molecules along or perpendicular to the principal diffusion direction and can help interpret FA findings [9].

### 2.4. Statistical Analyses

To test our hypothesis, we used a two-level analytic approach. False Discovery Rate (FDR *p* < 0.05) [43] was used to correct for multiple comparison. 

#### 2.4.1. Level 1: Primary Hypothesis Testing

The contribution of 33 pre-scan variables (including demographic, psychiatric comorbidities, and pre-scan longitudinal variables; for additional information, see Appendix A) to our main outcome (mean FA in five white matter tracts of interest—FMIN, left CG, right CG, left UNC, and right UNC) was assessed for feature selection using Least Absolute Shrinkage and Selection Operator (LASSO) regression within the GLMNET package [44,45]. 

Available in R [46], GLMNET fits a generalized linear model via penalized maximum likelihood using LASSO (alpha = 1) penalties [44,45]. The GLMNET algorithms use cyclical coordinate descent methods for solving penalized regression models and produce paths of solutions (tuning parameter lambda) [44,45]. GLMNET then uses cross-validation to identify the optimal lambda. The prediction error associated with alpha (the LASSO penalty) and lambda (the tuning parameter) is used to select the model with the least error and to avoid overfitting [44,45]. Non-zero coefficients characterize significant predictors and higher absolute coefficients reflect a higher contribution to the model. The multi-response Gaussian family (mgaussian) was used to reduce the dimensionality of the data and focus on those variables that best explained our outcomes of interest. 

Linear regressions were used to assess the effects of the variables identified by the LASSO model on the mean FA of each tract of interest (FMIN, left CG, right CG, left UNC, and right UNC) [46,47].

Focal abnormalities in tract of interest. For the tracts in which findings survived FDR correction, nodal FA values (100 consecutive nodes) were used to decide whether the tract abnormalities were focal or widespread. The cluster forming threshold was set to 10 contiguous nodes. In cases of 10 or more consecutive nodes in the same tract surviving FDR correction, mean FA was derived to reflect these focal abnormalities in further analyses. Mean AD and RD were also extracted from node clusters to help further interpret main FA findings.

Relationship with symptoms at scan. Pearson or Spearman correlations, as appropriate, were used to explore the relationships between mean FA of focal abnormalities identified in white matter tracts of interest and clinical symptoms collected at scan. 

#### 2.4.2. Level 2: Secondary Hypothesis Testing

To identify predictors of recurrence of depressive episodes one-year post-scan, we used LASSO regression in GLMNET (binomial family). Here, we examined the contribution of clinical, demographic, and neuroimaging variables to post-scan BD groups (Depressed BD; Non-depressed BD). The clinical and demographic variables included in Level 1 analysis were included in these analyses in addition to variables collected at scan (13 dimensions of symptoms and 3 classes of psychotropic medications). Of note, neuroimaging predictors were divided into two categories: (1) mean FA of node clusters identified in Level 1 analysis (representing white matter segments affected by pre-scan variables), and (2) mean FA of segments or tracts not associated with pre-scan variables. The described categories allowed us to assess the contribution of all regions of white matter tracts of interest in the prediction of depressive episodes. If no node cluster was identified in Level 1 analysis, the mean FA of the entire tract was entered as the predictor.

Variables selected by LASSO regression were imported to SPSS V24.0. Logistic regression analyses were used to identify the Adjusted Odds Ratio (AOR) associated with each selected variable. Receiver Operating Characteristic (ROC) curve and Area Under the Curve (AUC) were used to assess the diagnostic ability of the selected model [48].

Between-group comparisons. Analyses explored the group differences between Depressed BD, Non-depressed BD, and HC on mean FA of neuroimaging variables selected in Level 2 analyses. Given the known effect of age [49,50,51] and sex [52] on white matter, these variables were included as covariates. 

#### 2.4.3. Exploratory Analyses

Additional analyses examined if Depressed BD showed FA abnormalities in tracts or clusters not associated with the recurrence of depressive episodes one year after scan when compared to Non-Depressed BD and HCs. Given the known effect of age [49,50,51] and sex [52] on white matter, these variables were included as covariates. 

Given the known effect of movements on dMRI measures, the main effect of average translation and rotation was also explored in all node clusters identified in Level 1 and Level 2 analyses.

## 3. Results

### 3.1. Level 1: Primary Hypothesis Testing

LASSO regression in GLMNET revealed that four variables were associated with lower FA in the white matter tracts of interest: (1) age at scan, (2) being Caucasian, (3) number of depressive episodes during childhood/adolescence and (4) percentage of time experiencing syndromic depression. Coefficients are detailed in Table 2. 

Regression analyses revealed a significant effect of the variables identified in the LASSO model in three out of five white matter tracts of interest: 1. FMIN (F = 4.3, *p* = 0.003, FDR *p* = 0.009), 2. Left CB (F = 3.4, *p* = 0.014, FDR *p* = 0.024), 3. Right CB (F = 4.8, *p* = 0.002, FDR *p* = 0.009), 4. Left UNC (F = 1.6, *p* = 0.180, FDR *p* = 0.180), and 5. Right UNC (F = 1.7, *p* = 0.166, FDR *p* = 0.180). The FA variability in the FMIN was best explained by the number of depressive episodes during childhood/adolescence while the FA variability in both left and right CB was best explained by percentage of time experiencing syndromic depression during the study (Appendix A).

Focal abnormalities in tracts of interest. Tract-profile analyses revealed that the main effects were focal: in the middle left (Size= 11 nodes, F = 4.4, *p* = 0.003, FDR *p* = 0.015) and middle right (Size= 10 nodes, F = 3.1, *p* = 0.021, FDR *p* = 0.039) portions of the FMIN and in the anterior portions of the left (Size= 44 nodes, F = 4.7, *p* = 0.002, FDR *p* = 0.029) and right (Size= 53 segments, F = 7.0, *p* < 0.001, FDR *p* = 0.013) CB (Figure 2). There was no association between the clinical model and AD or RD in these node clusters (Appendix A). 

Relationship between focal abnormalities and symptoms at scan. Lower FA in the FMIN middle left node cluster was correlated with higher ALS, MASQ90 General distress depressive, MASQ90 General distress mixed, and STAIY Trait scores. Lower FA in the other three node clusters (FMIN middle right cluster, left CB anterior cluster, and right CB anterior cluster) was correlated with higher MASQ90 General distress depressive and STAIY Trait scores. There was no correlation between nodes FA and other symptoms at scan (Appendix A). 

### 3.2. Level 2: Secondary Hypothesis Testing

LASSO regression in GLMNET identified four predictors for the recurrence of depression post-scan: (1) BIS total score, (2) FMIN middle right cluster (one of the regions associated with pre-scan variables), (3) percentage of time experiencing syndromic depression, and (4) right CB posterior cluster mean FA (segment of the right CB not associated with pre-scan variables). Coefficients are detailed in Table 2. 

Logistic regression models revealed that higher BIS total score (AOR = 2.42, *p* = 0.010) and higher pre-scan percentage of time experiencing syndromic depression (AOR = 3.33, *p* = 0.007) were associated with increased risk of recurrence of depressive episodes one year after scan. The BIS total score corresponds to the sum of three second-order factors (attentional, motor, and non-planning impulsiveness) [37]. Further analyses using these three factors revealed that only the attentional factor (AOR = 2.8, *p* = 0.005) was able to predict recurrence of depression (for additional information, see Appendix A). By contrast, higher FA in FMIN middle right cluster (AOR = 0.43, *p* = 0.017) and right CB posterior cluster (AOR = 0.50, *p* = 0.032) were protective against new episodes one year after scan (Table 3). The model combining all four predictors showed an AUC of 0.85 (Appendix A).

Between-group comparisons. Participants in the Depressed BD group showed lower FA in FMIN middle right cluster (F = 6.9, *p* = 0.011, FDR *p* = 0.022) and in the Right CB posterior cluster (F = 5.0, *p* = 0.031, FDR *p* = 0.031) in comparison to Non-depressed BD group (Figure 3). When compared to the HC group, the Depressed BD group also showed lower FA in FMIN middle right cluster (F = 7.8, *p* = 0.007, FDR *p* = 0.014) and in the Right CB posterior cluster (F = 4.6, *p* = 0.037, FDR *p* = 0.037). Non-depressed BD and HC groups did not show between group differences in these regions (FMIN middle right cluster, F = 0.1, *p* = 0.702, FDR *p* = 0.908; right CB posterior cluster, F < 0.1, *p* = 0.908, FDR *p* = 0.908). 

### 3.3. Exploratory Analyses

Depressed BD did not show any FA difference in tracts or clusters not associated with the recurrence of depressive episodes one year after scan when compared to Non-Depressed BD and HCs (Appendix A). 

There was no correlation between translation and rotation and FA in tracts or node clusters included in Level 1 and 2 analysis (Appendix A).

The model identified in Level 1 analysis was associated with lower FA in the forceps major (F = 3.9, *p* = 0.007, FDR *p* = 0.077) and anterior thalamic radiation bilaterally (left, F = 3.2, *p* = 0.019, FDR *p* = 0.105; right, F = 2.6, *p* = 0.046, FDR *p* = 0.169). However, these findings did not survive FDR correction (Appendix A). 

## 4. Discussion

This study aimed to promote a better understanding of the neural mechanisms of BD by examining the effect of previous mood symptoms on white matter tracts that have been largely implicated in the pathophysiology of BD and the extent to which they can contribute to the prediction of recurrence of future depressive episodes. 

Our findings indicate that experiencing more depressive episodes during childhood/adolescence and more syndromic depressive symptoms over the years were associated with lower fiber collinearity in focal regions of the FMIN and CB. These effects were mainly present in the middle segments of left and right portions of the FMIN and in the anterior segments of the CB. Given that AD and RD were not affected by the pre-scan model, these FA abnormalities are probably the result of additive changes in axial and radial directionalities. Older age at scan was also associated with lower FA in these regions, a finding consistent with previous studies that showed that aging is associated with lower fiber collinearity [53,54]. The FMIN connects contralateral prefrontal regions [55] while the CB constitutes the dorso-limbic pathway, interconnecting structures of the limbic system involved in reward, default mode network, and emotional regulation [56]. Both tracts play a key role in the pathophysiology of BD, with BD participants showing lower FA when compared to healthy participants [11,15,16,17,18,26,57,58]. Our findings revealed that lower fiber collinearity in these tracts could reflect the cumulative effect of years of depressive symptoms. In BD, incidence of mood symptoms has been associated with progressive and cumulative neuronal damage [59,60,61]. This process, called neuroprogression, is associated with worsening of BD and progressive clinical deterioration over the years [59,60,61]. It is hypothesized that persistence and recurrence of mood episodes could lead to permanent alterations in neuronal circuits through increased oxidative stress and inflammation [62]. Consistent with this hypothesis, a cross-sectional study showed that lower FA was associated with higher levels of lipid peroxides (metabolite generated by oxidative stress) across BD and HCs samples and that BD participants showed higher levels of this metabolite in comparison to controls [13]. In addition, previous studies have also shown that individuals with BD experience larger decline in gray matter volume over time in comparison to HCs and that these reductions were associated with the number of mood episodes and illness duration [63,64,65]. However, to the best of our knowledge, this is the first BD study to examine the effects of decades of mood symptoms on white matter architecture/integrity and to show a potential cumulative effect of experiencing more depressive episodes/symptoms over time.

Further analyses revealed that higher fiber collinearity in regions of the FMIN and CB was protective against the recurrence of depressive episodes one year after scan. These regions were the FMIN middle right cluster and the right CB posterior cluster. Both white matter regions were also able to differentiate Depressed BD from Non-depressed BD and HCs. As noted above, lifetime depression was associated with lower FA in the FMIN cluster, suggesting that the cumulative effect of years of BD might cause structural abnormalities in this region and increase the risk for recurrence of future depressive episodes. These findings are in line with the neuroprogression of BD [59,60,61]. However, while the anterior portion of the CB was associated with lifetime depression, only the posterior CB was associated with future recurrence of depression. The CB plays a key role in the pathophysiology of BD and abnormalities in this tract have been associated with a familial vulnerability for psychopathology in individuals at risk for BD [66]. Our findings suggest that there is a distinction between regions affected by neuroprogression effects and neural prediction of future depressive episodes in this tract. The CB has a complex architecture, including both short and long associative fibers that connect frontal, parietal, and temporal lobes [67,68]. This structural distinction might inform potential neural targets for neural modulation techniques, such as transcranial Direct Current Stimulation (tDCS), which has shown promising effects in reducing depressive symptoms [69]. We have also shown that a combination of clinical and neuroimaging predictors showed high predictive power to differentiate BD participants with depressive episodes one year after scan from those without recurrence. In addition to the FMIN middle right cluster and the posterior CB FA, predictors included a higher percentage of syndromic depressive symptoms before scan and a higher BIS total score at scan (mainly from the attentional component). Consistent with our findings, studies have shown that recurrence of depressive episodes is directly associated with history of depression, number of previous episodes, and attentional deficits [70,71,72].

There were limitations in this study. (1) Although we evaluated the impact of up to 18 years of clinical symptoms on white matter integrity, this study did not include dMRI data before the onset of BD symptoms. To address this limitation, we included dMRI data of healthy controls as a normative reference. However, this approach cannot rule out the possibility that these abnormalities existed before study entry. Future naturalistic studies should include neuroimaging data before BD onset and prospectively follow-up symptoms over the course of many years. (2) We acknowledge that the retrospective recollection of mood symptoms is challenging and that recall bias can affect the retrospective self- and parental reporting of symptoms. (3) Perhaps due to ongoing pharmacological treatment, the recurrence of (hypo)manic symptoms was infrequent after scan. Therefore, this study focused only on the prediction of depressive symptoms with 36% of the participants experiencing depression up to one year after the scan. In line with our findings, other studies that focused on distinct BD samples found similar recurrence of depression in the first two years (34.7%) and showed that (hypo)manic episodes were less common (13.8%) during this time and happened mostly after the first year [73]. Future neuroimaging studies should evaluate the association between the cumulative effects of lifetime depression in BD and the recurrence of mood episodes beyond the first year of follow-up. (4) Our findings showed that being Caucasian was associated with lower FA in white matter tracts of interest. However, this sample included a high proportion of Caucasians. Future studies should include a more diverse BD sample to properly address the effects of ethnicity on white matter. (5) The study only accounted for the presence of medications (YES/NO) at the time of the scan, but did not evaluate the effects of different dosages, duration, blood levels (when appropriate), polypharmacy, and adherence to treatment. The relationship between the use of medications and course of illness in a naturalistic study is complex and beyond the scope of this paper.

## 5. Conclusions

To our knowledge, this is the first study to demonstrate the cumulative effects of lifetime depression in the FMIN and CB and to show their roles on the recurrence of future depression in BD. These findings further our understanding of the neural mechanisms of BD and inform on the importance of accounting for long-term symptomatology in mood spectrum disorders, which can elucidate the contribution of different neural regions/circuits and should promote the development of new ad hoc preventive treatments (e.g., neural modulation) for BD.

## Figures and Tables

**Figure 1 jcm-11-03432-f001:**
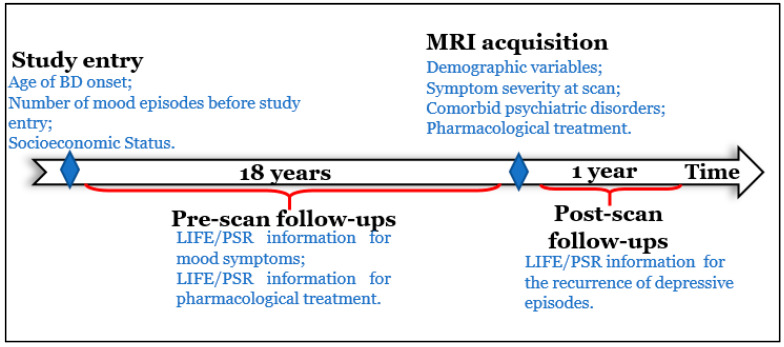
Study timeline. Figure 1 shows the timeline and assessments of the study. There are four main sources of information in this study: (1) Study entry, (2) pre-scan follow-ups, (3) MRI acquisition, and (4) post-scan follow-ups. At study entry, information regarding age of BD onset, number of mood episodes before study entry (during childhood and adolescence), and socioeconomic status was collected. Pre-scan follow-ups assessed severity and frequency of depression, mania, and hypomania using LIFE. These symptoms were assessed on a weekly basis for the interval between follow-ups. Information regarding long-term exposure to medications (antidepressants, mood stabilizers, antipsychotics, stimulants, and benzodiazepines) was also assessed. At MRI visit, all BD participants were assessed for symptom severity, comorbid psychiatric disorders, and current pharmacological treatment in addition to the neuroimaging acquisition. A total of 70 participants had study-entry, pre-scan follow-up, and usable neuroimaging data. Neuroimaging data and cross-sectional assessments were also acquired for 39 HCs. Post-scan follow-ups used LIFE/PSR to identify the recurrence of depressive episodes. Depressive episodes were defined as two or more consecutive weeks of PSR ≥ 5 for depression. Fifty-two participants showed post-scan follow-up data. Abbreviations: BD, Bipolar Disorder; LIFE, Longitudinal Interval Follow-up Evaluation; PSR, Psychiatric Status Ratings; HC, Healthy Controls.

**Figure 2 jcm-11-03432-f002:**
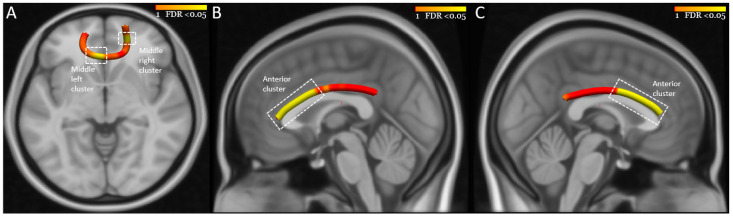
Neural correlates of previous bipolar illness. (**A**–**C**) show the node clusters in the FMIN, left CB, and right CB. Tracts were reconstructed using Tracula. The background is the standard MNI-152 1 mm brain. The red-yellow color bar represents the range of *p* values used in node-wise statistics after FDR correction. Abbreviations: FMIN, Forceps Minor; CB, Cingulum Bundle.

**Figure 3 jcm-11-03432-f003:**
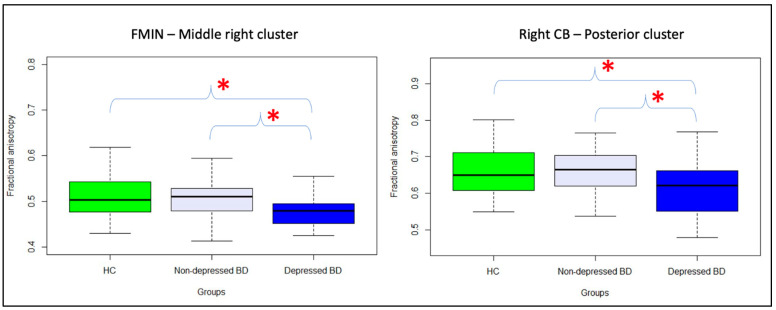
Neural predictors of future recurrence of depressive episodes in BD. Boxplots in panels A and B show the group difference upon FA in 39 HCs (green color), 33 BD participants without recurrence of depressive episodes after scan (Non-depressed BD, light blue color), and 19 BD participants who showed recurrence of depressive episodes after scan (Depressed BD, blue color). Braces and asterisks show *p*-values that survived FDR correction. Abbreviations: FMIN, Forceps Minor; CB, Cingulum Bundle.

**Table 1 jcm-11-03432-t001:** Sample characteristics.

Characteristics	Total (N = 109)	HC (N = 39)	BD (N = 70)	t(107) or χ^2^	*p* Value ^a^
Age (years), mean (SD)	26.2 (4.0)	25.4 (4.5)	26.3 (3.9)	−1.2	0.254
Sex, No. (%)					
Men	52 (47.7%)	17 (42.6%)	35 (50.0%)	0.4	0.521
Women	57 (52.3%)	22 (56.4%)	35 (50.0%)		
Educational Level, ^b^ No. (%)					
Higher	40 (36.7%)	26 (66.7%)	14 (20.0%)	23.5	**<0.001**
Lower	69 (63.3%)	13 (33.3%)	56 (80%)		
Handedness, No. (%)					
Left	17 (15.6%)	6 (15.4%)	11 (15.7%)	<0.1	0.964
Right	92 (84.4%)	33 (84.6%)	59 (84.3%)		
Race, No. (%)					
Caucasian	72 (66.1%)	20 (51.3%)	52 (74.3%)	5.9	**0.015**
Non-Caucasian	37 (33.9%)	19 (48.7%)	18 (25.7%)		
Employment Status, No. (%)					
Employed	64 (58.7%)	14 (35.9%)	50 (71.4%)	36.4	**<0.001**
Unemployed	15 (13.8%)	1 (2.6%)	14 (20.0%)		
Full-time student	30 (27.5%)	24 (61.5%)	6 (8.6%)		
Clinical characteristics at-scan, mean (SD)					
HDRS	6.8 (6.6)	1.4 (1.6)	9.8 (6.5)	−10.3	**<0.001**
YMRS	2.6 (3.0)	0.4 (0.9)	3.9 (3.1)	−8.8	**<0.001**
BIS	60.2 (13.1)	51.8 (10.2)	64.9 (12.2)	−5.7	**<0.001**
ALS	40.3 (36.1)	17.8 (18.7)	52.9 (37.5)	−6.5	**<0.001**
SSS	17.7 (4.5)	18.7 (5.9)	17.2 (6.2)	1.3	0.206
STAIY State total	34.6 (10.8)	28.4 (8.9)	38.1 (10.3)	−4.9	**<0.001**
STAIY Trait total	37.1 (11.1)	30.0 (7.7)	41.1 (10.8)	−6.2	**<0.001**
MASQ90 Anhedonic depression	57.1 (14.4)	50.8 (12.0)	60.5 (14.5)	−2.5	**0.001**
MASQ90 Anxious arousal	20.8 (7.1)	17.9 (1.4)	22.5 (8.4)	−4.4	**<0.001**
MASQ90 Loss of interest	12.7 (5.6)	9.7 (1.7)	14.4 (6.3)	−5.8	**<0.001**
MASQ90 General distress—Depressive	18.5 (8.6)	14.2 (2.6)	20.9 (9.8)	−5.4	**<0.001**
MASQ90 General distress—Anxious	16.4 (6.1)	13.4 (2.9)	18.0 (6.8)	−5.0	**<0.001**
MASQ90 General distress—Mixed	27.8 (11.0)	20.2 (3.7)	32.0 (11.4)	−7.9	**<0.001**

Abbreviations: HC, Healthy Controls; BD, Bipolar Disorder; HDRS, Hamilton Rating Scale for Depression; YMRS, Young Mania Rating Scale; BIS, Barratt Impulsiveness Scale; ALS, Affective Lability Scale; SSS, Sensation-Seeking Scale; STAIY, State-Trait Anxiety Inventory; MASQ90, Mood and Anxiety Symptom Questionnaire. ^a^ *p* values ≤ 0.050 are reported in bold characters. ^b^ Educational level was defined as higher (college degree and above) and lower (high school or some college).

**Table 2 jcm-11-03432-t002:** LASSO coefficients.

**Level 1 Analysis—Pre-Scan Predictors of White Matter Fiber Collinearity at Scan**	
**Variables**	**Coefficient**
Age at scan	−0.02
Caucasian	−0.09
Number of depressive episodes during childhood/adolescence	−0.05
Percentage of time experiencing syndromic depression	−0.09
**Level 2 Analysis—Predictors of Recurrence of Depressive Episodes One Year after Scan**	
**Variables**	**Coefficient**
BIS total score	0.17
FMIN middle right cluster	−0.04
Percentage of time experiencing syndromic depression	0.34
Right CB posterior cluster	−0.02

Abbreviations: BIS, Barratt Impulsiveness Scale; FMIN, Forceps Minor; CB, Cingulum Bundle.

**Table 3 jcm-11-03432-t003:** Logistic regression results—Predictors of recurrence of depressive episodes one year after scan.

Variables	B	Wald	*p* Value ^a^	AOR	95% CI	
BIS total score	0.88	6.65	**0.010**	2.42	1.24	4.72
Percentage of time experiencing syndromic depression	1.20	7.30	**0.007**	3.33	1.39	7.99
FMIN middle right cluster	−0.84	5.66	**0.017**	0.43	0.21	0.86
Right CB posterior cluster	−0.70	4.58	**0.032**	0.50	0.26	0.94

Abbreviations: BIS, Barratt Impulsiveness Scale; FMIN, Forceps Minor; CB, Cingulum Bundle. ^a^
*p* values ≤ 0.050 are reported in bold characters.

## Data Availability

The data may be available upon reasonable request.

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
