# Peer review of "White Matter Correlates of Early-Onset Bipolar Illness and Predictors of One-Year Recurrence of Depression in Adults with Bipolar Disorder"

_jcm, 2022, doi:10.3390/jcm11123432_

Round 1

Reviewer 1 Report

In this study, the authors used global probabilistic tractography to identify neural correlates of previous Bipolar Disease (BD) and to predict one-year recurrence of depressive episodes. Their main results show that more recurrent depressive episodes during childhood/adolescence and syndromic depression were associated with reduced fractional anisotropy (FA) in important regions for BD, and that higher FA in specific regions was protective against depressive episodes.
The study is interesting and has good impact (including clinical impact), highlighting the impact of depression in BD symptomatology. Also to be appreciated is the focus on white matter, whereas most studies on BD have looked at gray matter and functional imaging. The methodology used seems to me to be consistent with the experimental design, the sample size seems adequate, and the analyses were adjusted for multiple comparisons. Some suggestions for improving the manuscript follow.

Title: I would rephrase "Bipolar Adults" with "Adults with Bipolar Disorder" instead.

Line 55-56: The authors should be more cautious in their statement. Having no MRI data at the onset of BD, it is not properly correct to speak about cause-consequence. I would suggest rephrasing.

Line 90: The authors state that healthy controls had no Axis-I diagnosis. How was this verified? Were they subjected to a clinical interview?

I think Figure S1 should be in the main manuscript, this would make the experimental design of the whole study clearer.

Line 183: The authors include the Barratt Impulsiveness Scale (BIS) in the measures, but the role of impulsivity in BD was not adequately introduced. Considering that an effect of impulsivity emerges in subsequent analyses, I would suggest better emphasizing the importance of impulsivity in BD in the introduction.

Line 231 (As well as, for example, at line 238) I would use italics for the headers of these subsections.

Figure 2: It would be desirable to report boxplots instead of bar plots so that the distribution of variables among groups can be shown.

Reviewer 2 Report

This study used lobal probabilistic tractography to identify neural 19 correlates of previous BD illness and the extent to which these can help predict one-year recurrence 20 of depressive episodes. The design of primary hypothesis testing, secondary hypothesis testing and exploratory analyses were very clear and rational. Overall, this is a well-done study with very interesting findings. Some suggested edits:

1.     Diffusion Magnetic Resonance Imaging is not an accuracy term, it can be diffusion weighted imaging, Diffusion tensor imaging, or Diffusion Basis Spectrum Imaging et.al. It looks like DTI in the current study, but the author may make this clear in the main text.

2.    Why Linear regressions and LASSO model was chosen should be more clear in the introduction and method. I think there might be more complexity mode can be applied to test the predict ability of current finding, such as logistics regression or SVM.

Reviewer 3 Report

The references cited in the article are update to 2022 and so they are recent. 

The sample is sufficient and equally distributed between BD and HC, but there is a difference regarding the level of instruction between the two groups: in fact, BD sample has more subjects with lower schooling respect to HC.  

The instrument used are favorite in guidelines but are all indirect measure (questionnaire). For example, Barratte Impulsiveness Scale is a gold standard to measure the impulsiveness in bipolar patients but it is an indirect measure for cognitive planning. For these reasons, the authors could think to insert also a direct assessment of cognitive functioning in future research. 

The limits are exposed very clearly and open future possible ways in this research field.

Author Response

We thank the reviewer for their time and thoughtful comments on our manuscript entitled “White Matter Correlates of Early-Onset Bipolar Illness and Predictors of One-Year Recurrence of Depression in Adults with Bipolar Disorder”. We plan to expand our models in future research by including objective measures of cognitive functioning and evaluating the association between these measures and white matter.